# Exploring the Value Co-Creation of Cultural Creative Hotels: From the Perspective of Social Innovation

**Mai-Rong Wang [1] and Chun-Liang Chen [2],***

[1] Crafts & Design Department, National Taiwan University of Arts, Banciao Dist., New Taipei City 22058, Taiwan; indingo.wang@gmail.com

[2] Graduate School of Creative Industry Design, National Taiwan University of Arts, Banciao Dist., New Taipei City 22058, Taiwan

* Correspondence: jun@ntua.edu.tw

**Abstract:** Social innovation has emerged as a transformative force in businesses, particularly in the context of cultural and creative hotels. This study aims to explore the relationship between social innovation and value co-creation in cultural and creative hotels, compare the intrinsic characteristics of social innovation and value co-creation within these hotels, and investigate the key factors driving social innovation in this sector. Employing a qualitative research methodology based on the theory of social innovation, this paper examines the process of value co-creation and analyzes the three key drivers within the social innovation ecosystem: establishing interdependence and identity among organizations; enhancing cognitive and value exchanges between organizations; and generating consensus through the role transformation of participants. The findings suggest that the key drivers of the social innovation ecosystem not only enhance the innovation capabilities of businesses but also motivate them to collaboratively create mutually beneficial and symbiotic value.

**Keywords:** service ecosystem; innovation ecosystem; social innovation; value co-creation; service-oriented logic

## 1. Introduction

In June 2023, the government initiated the "Social Innovation Action Plan 2.0", focusing on four major strategies aimed at establishing public–private partnerships, strengthening the autonomous operation and development of social innovation organizations, finding the most suitable solutions, and expanding the influence of sustainable society. In the post-pandemic era, enterprises need to approach the changes in the market environment from their own perspective. The impact of value co-creation on the sustainable development of enterprise organizations cannot be ignored. Therefore, exploring organizational value co-creation activities from a social innovation perspective is particularly important. Social innovation is defined as the process by which enterprises create socio-economic value through value creation, altering corporate culture or corporate structures, thereby enhancing market competitiveness [1].

In recent months, Taiwan Trends Research Company has reported that the epidemic has had a serious impact on the hotel and service industries [2], especially those involved in cross-border tourism, leading to business cuts and layoffs. Since the lifting of border restrictions on 13 October 2022, the global tourism industry has recovered. Though international tourists have not yet returned, the shortage of manpower in the tourism industry remains, especially in the hotel industry (hereinafter referred to as cultural creative hotels, tourism). A cultural creative hotel can be defined as a for-profit organization that incorporates local culture and unusual design elements in its service offerings, which include accommodation and experiential activities with cultural and recreational value [3]. Despite the growth of the tourism market, it is facing fierce market competition. Hotel

operators need to consider how to establish value co-creation strategies through social innovation to address social challenges. Therefore, promoting value co-creation in a competitive environment makes social innovation a key to maintaining competitiveness.

From the perspective of the service ecosystem, innovation is influenced by geographical environment and human factors, and it is necessary to propose new and improved services to meet the constantly changing consumer demands of the market [4]. This study focuses on the role of cultural creative hotels in value co-creation in the social innovation ecosystem, which involves multiple domains, including the industrial environment, partnerships, and the innovation ecosystem. Many scholars have proposed theoretical explorations in this research topic. Ahmadi et al. [5] investigated the factors and frameworks that impact social innovation after the pandemic. Soni et al. [6] analyzed the application of social innovation and digital communications technology in enterprises, proposing sustainable development strategies. Singh et al. [7] argued that the public sector should propose specific actions for social innovation co-creation. However, research on cultural creative hotels guided by the social innovation ecosystem remains lacking.

Therefore, this study explores the value co-creation strategy mechanism of cultural creative hotels from the perspective of the social innovation ecosystem and how to achieve the coexistence of interests. The purposes of this study are as follows:

(1) Explore the social innovation and value co-creation of cultural creative hotels.
(2) Compare the social innovation and value co-creation characteristics of cultural creative hotels.
(3) Explore the key elements driving the social innovation ecosystem of cultural creative hotels.

## 2. Literature Review

### 2.1. Service-Dominant Logic and Service Ecosystem

Vargo and Lusch [8] combine the concept of the ecosystem with service-driven logic and propose the argument of the "service ecosystem". Vargo and Lusch [9] argue that service-driven logic regards all commercial and economic activities as "services", defining services as the use of resources to create the interests of other actors while also serving as the foundation of all economic exchanges. Vargo and Lusch [9] argue that from the perspective of service-driven logic, when exploring market segmentation, roles are not divided based on the identities of suppliers or demanders. Instead, all participants are viewed as "actors", who simultaneously play the role of resource integration, that is, providing services while also obtaining services. They emphasize that all parties involved in the market have a dual identity of both suppliers and demanders in their actions. Service-oriented logic emphasizes that enterprises should place customer needs at the core and achieve the goal of value co-creation by providing valuable services [10].

Frow and Payne [11] argue that service-oriented logic involves multiple stakeholders, and different stakeholders work together to create a value coordination mechanism. Companies can handle management issues in a more comprehensive and integrated manner. This new organizational form makes companies more flexible, able to quickly respond to market changes, meet customer needs, and create sustainable competitive advantages. The broad definition of stakeholders, according to [12], refers to any individual or group that affects the achievement of organizational goals or is influenced by any organizational activity. "Value proposition" is also an important concept in service-oriented logic. Chandler and Lusch [13] observe that value propositions can attract customer engagement while involving stakeholders throughout the entire service system network. Therefore, the value proposition connects the enterprise and stakeholders. This echoes the viewpoint of [13], which is that the enterprise proposes a value proposition, involves relevant stakeholders in the process of value creation, actively creates value that meets customer needs, and enables stakeholders in the service ecosystem to participate in value creation and play a key role. Vargo and Lusch [9] propose that service ecosystem management policies include resource decision-making, creation and integration, the establishment of shared

beliefs and values, attention to resource exchange when norms arise, and the consideration of institutional arrangements between a wider range of participants and resource systems.

In sum, the development of service ecosystem management strategies must consider consensus, rules, and the complexity of the entire resource system. If sustainable management strategies are comprehensively developed within an organization, they can meet the needs and expectations of society. Wennerholm [14] suggests that value co-creation occurs when suppliers and customers jointly invest in the service system for knowledge and information exchange. Only when the service system masters the information of customers or competitors can its own service innovation develop. If enterprises want to maintain their functionality and vitality in the value network, they need to implement service-oriented logical concept thinking in service ecosystem organization, break traditional forms and rules, introduce new information to reconstruct resources, strengthen the tightness of the enterprise value network, and implement competitive value propositions and learning opportunities [15].

*2.2. Value Co-Creation*

In the latest research on value co-creation, Ehret and Wirtz [16] advocate that businesses can reduce the uncertainty generated by consumers in the co-creation process by presenting clear value propositions and clarifying their substantive value significance to consumers. When consumers participate in production activities, the boundary between production and consumption roles gradually becomes blurred, which means that the realization of value co-creation depends on the interaction and resource integration among all parties [17]. Vargo and Lusch [18] propose that the development of the market must be composed of the following components: service exchange, resource integration, value co-creation, and value creation. They emphasize that value co-creation is a cyclical system, and the cyclical process must have certainty, including the participation of stakeholders, resource integration, and service exchanges, to form a service system.

Vargo and Lusch [9] contend that in a service ecosystem, collaboration and coordination among different participants can inspire new strategies and promote the continuous development of the system through institutionalization and mechanism arrangement. At the same time, it also helps to evaluate market mechanisms more comprehensively and accurately. Institutionalization makes thinking integration more targeted and directional. Wennerholm [14] believes that co-creation refers to the joint investment of suppliers and customers. For example, by introducing new information technology, enterprises can integrate various technological resources, improve information service systems, narrow geographical barriers, and bring service providers closer together. To achieve value co-creation in the service system, service providers must be responsible for proposing clear value propositions and value co-creation, as only in practice can we achieve a balanced relationship between the two.

In summary, value Co-creation within the operation of cultural and creative hotels involves customer participation in designing customized innovative services, providing service demands and improvement suggestions, or participating in social interactions. This enhances the service experience and increases satisfaction, where consumers, users, or customers actively partake in the service process to co-create value.

*2.3. Social Innovation*

2.3.1. Social Innovation and Strategic Alliance

Avelino et al. [19] cited Westley's definition of social innovation in his conference speech in transformative social innovation, which refers to the persistence, scale, and transformative impact of successful social innovation. Any product, planning process, project, or platform that continues to challenge the system over time can help introduce a broader definition of the social system (such as faith, authority, conventions, and

resources). Saka et al. [1] observed that the definition of social innovation (CSI) at the enterprise level refers to improving a company's competitiveness by changing its structure, strategy, and culture while creating economic and social value.

Emami et al. [20] argued that in strategic alliances, companies often rely on the key abilities of their participants during the start-up phase, especially in the development of new technologies. Key abilities are particularly important, and enterprise innovation requires the ecosystem to provide new information and services to jointly create new value for customers. In order to encourage service ecosystem participants to actively participate in the value co-creation process, efforts must be made to ensure a clear vision for value co-creation. Participants in each ecosystem of the alliance establish new relationships and, most importantly, the contribution between partners and participants. The greater the contribution in the value creation process, the greater the innovation ability.

Babu et al. [4] argued that strategic alliances promote the establishment, participation, and evolution of social innovation in service ecosystems, ultimately achieving value co-creation through sustained social innovation. Therefore, Babu et al. [4] noted that strategic alliances between organizational departments can allow them to fully leverage their respective knowledge and expertise, connect partners, strengthen the establishment of alliance networks, and promote social change to facilitate broader social participation, especially to address unmet needs in the market. This can provide more comprehensive services. This spiral and repetitive relationship constitutes the driving force of value creation and strategic alliances in the service ecosystem, while strategic alliances in turn sustain social innovation. The interaction between organizations is not carried out in a traditional structured manner but rather reflects more informal and flexible cooperative network relationships.

### 2.3.2. Social Innovation and Service Ecosystem

Anheier et al. [21] argue that social innovation is purposeful and strategic social innovation, which not only benefits society but also enhances individuals' ability to take action. They view social innovation as a phenomenon of social cohesion. The fundamental spirit of social innovation is that organizations must engage in imagination and creation, learn from failure, and solve the final problem. The key is to clarify these roles and their respective goals and how to interact and cooperate with others in the process. Actors who promote innovation must also have the capacity to innovate and contribute to society [22].

In sum, social innovation refers to the process of introducing new ideas and methods or applying technological models in social and economic fields, solving existing problems and improving social conditions to promote the creation of new value. Avelino et al. [19] suggest that service ecosystems can collaborate with participants in social innovation and through the integration of change and innovation, enhance their ability to change through the process of evolution. This innovation is not the efforts of a single organization or individual but is built on the basis of multi-party cooperation and interaction, forming an interconnected system that stimulates the innovative power of services and organizations. The interactions in service ecosystems are variable dynamic systems, and different scenarios evolve over time, enabling ecosystems to regulate various environmental changes and generate innovative collaborative cooperation [23].

In summary, the goal of social innovation for cultural and creative hotels is to foster inclusive growth and community engagement through innovative solutions. For example, cultural and creative hotels may support local artists by offering creative workshops, which not only promote local cultural development but also bring economic benefits to the community.

### 2.3.3. Social Innovation and Value Co-Creation

Lusch and Nambisan [23] state that participants are potential value co-creators, and this ecosystem draws on the concept of digital value networks established by information technology (IT), integrates knowledge in resource networks, and triggers service exchange

to generate innovative capabilities through the flow of shared resources. Participants, whether individuals or organizations, participate in a way governed by rules on the service exchange platform without the need for control but coordinate with each other under self-restraint [8].

In sum, service ecosystems often exhibit characteristics of autonomous operation and self-regulation in economic and social networks, while innovation not only involves inventing new things but also involves recombining or integrating existing elements. Therefore, Lusch and Nambisan [23] argue that service ecosystems can be seen as a new type of corresponding (A2A) structure that requires a service platform for network service exchange, on which all participants play the role of resource integrators to assist in resource exchange and improve resource density efficiency. This allows the network of participants in the value co-creation process to create new resources with different resources. Babu et al. [4] believe that the three concepts of value co-creation, the service ecosystem, and the service platform can serve as the fundamental categories of social innovation. In short, participants initiate innovation and stimulate value delivery abilities through interactive communication.

Following the trend of the times, digital networks are integrated into the operation of service ecosystem organizations across multiple domains, and value co-creation is driving the value proposition of social innovation. Lusch and Nambisan [23] proposed a service innovation theory perspective, which enables innovation to succeed and continue through information networks, resource flow, density creation, resource integration, and value co-creation among participants. Battisti et al. [24] proposed the interaction between different stakeholders, in which actors assist in social innovation and unleash value co-creation. At the same time, these interactions in turn form new co-creation service features, expanding the scope of social innovation beyond corporate activities. Therefore, Vargo et al. [25] argued that value co-creation, based on the ultimate goal of social innovation, is determined by the interaction between multiple stakeholders.

Social innovation refers to the three concepts of the service ecosystem, service platform, and value co-creation analyzed by [3], which appear in many studies. For example, Zahoor [26] contended that mutual trust among alliance partners enables information service platforms to generate better social network connections. Zhang et al. [27] proposed using platform cooperation alliances to achieve social innovation and solve the contradiction between value creation and redistribution through dynamic cooperation. Akter et al. [28] argued that when participants play a mediating role between organizations, it directly or indirectly affects the value co-creation performance in the quality of life. Roberts et al. [29] observed that value co-creation relationships are derived from the dimensions of interaction between different participants.

Saka et al. [1] argued that based on social innovation theory, strategic alliances are established on the concept of value co-creation, and the relationships between participants play a crucial role through collaborative cooperation, which can influence new innovation pathways in the field. Ahmadi et al. [5] contended that the joint participation of participants between departments in creating new products or services typically requires the establishment of a multi-party collaborative network to improve people's quality of life. Lind et al. [30] noted that in the context of service ecosystems, the success of social innovation depends on the interaction and sharing between different organizational main systems. Meister et al. [31] argued that social innovation in EU relief plans should be studied through action based on co-creation.

In summary, the innovation ecosystem, whether for internal organization or external units, involves the necessary integration and allocation of resources by cultural and creative hotels working together with participants or stakeholders, ultimately creating value. This includes interactions of knowledge, funding, technology, and markets, providing the development and implementation of new ideas. For cultural and creative hotels, being part of a regional social innovation ecosystem enables access to new creative resources and fosters cross-sector alliances. For instance, collaborations with local artists, cultural

organizations, and the community can co-deliver new value experiences and implement innovative concepts to advance value creation for each group.

In sum, with the ideas proposed in Section 2.3.1–2.3.3 (Social Innovation and Strategy Alliance, Social Innovation and Service Ecosystem, and Social Innovation and Value Co-creation), social innovation and strategic alliances, social innovation and service ecosystems, and social innovation and value co-creation are the theories of a number of scholars as a basic framework, using different constitutional perspectives to analyze, and will deconstruct the relationship between cultural creative hotels and value co-creation through the A2A network, resource flow, resource density, and resource integration in the social innovation ecosystem.

This study primarily explains the connotation and characteristics of the social innovation ecosystem in a conceptual and targeted manner. Although most scholars have adopted the interpretation of service ecosystems, there is less research on the co-creation of social innovation ecosystem values in cultural creative hotels. This article will expand from line to surface, develop a social innovation ecosystem, explore the key drivers of the social innovation ecosystem of cultural creative hotels, and present two case studies: The Place Taichung hotel [32] and the Chanyee Hotelday Hualien Mountain Knowledge (hereinafter referred to as The Place Hotel and Chanyee Hotelday) [33] as research objects. We observe the operational strategies of value activities in cultural creative hotels, explore how to extend and develop a value co-creation network for social innovation ecosystems both internally and externally, and analyze how each ecosystem integrates resources to discover key elements driving innovation.

## 3. Research Methods

### 3.1. Case Study Method

According to [34], case studies can be divided into three types: exploratory, descriptive, and explanatory (confirmatory), depending on the development of the research question in the theoretical context. The main purpose of explanatory case studies is to verify causal relationships, and the applied context is usually established when the theory is relatively mature. This indicates that researchers have a deep understanding of the research field, when new environmental factors appear, research predictions may differ, and explanatory case studies can be used to revise or extend existing theories. Therefore, the use of the explanatory case study method in this study is appropriate.

Combined with the reliability and validity concepts of [34] and the logic of replication, theoretical guidance may be established from case studies, emphasizing the inter-case and cross-case analysis of internal constructs in the case, and corresponding discussions of the literature. This article adopts a qualitative research method to use the researcher's perspective to shed light on practical phenomena (such as innovative experiential services in the actual operation of hotel operators), thus generating resonance [35]. Under the framework of qualitative research, practical phenomena are suitable for case studies [36] to achieve value co-creation by focusing on why or how phenomena occur and how to solve various problems, which is more conducive to the dialectical relationship between practice and theory. Based on this, qualitative case studies allow us to conduct an in-depth analysis of practical phenomena within a limited scope through case studies [37].

### 3.2. Case Selection

This study selected two cultural creative hotels in Taiwan with distinct geographical locations and cultural characteristics, Chanyee Hotelday (located in Hualien, Taiwan) and The Place Hotel (located in Taichung, Taiwan). These two regions offer different geographical environments. Hualien is renowned for its rich Indigenous culture, natural landscapes, and ecological resources, while Taichung is renowned for its diverse urban culture and arts. For their operational strategies and market positioning, cultural creative hotels adopt

themed designs with different cultural backgrounds, as different presentation methods will attract customers of different levels.

Eisenhardt [38] observed that theoretical sampling is a sample selection method based on research objectives and theoretical guidance, which can rationalize research and enrich the development of theory. Theoretical sampling should first clarify that a specific population helps to reduce external variability and ensure that the research results are credible within a specific range. Preliminary criteria for sample selection should be formulated based on the research objectives. These guidelines can ensure that the extracted cases meet the research objectives and theoretical needs. In fully explaining the construction and formation of the social innovation ecosystem, this study gradually confirms the theoretical saturation through theoretical sampling verification and can also present the significance of variables in actual situations [38], which is a commonly used strategy in case studies [34].

### 3.3. Data Collection and Analysis

The criteria for choosing cultural and creative hotels are based on their ability to showcase local cultural connections through design, service, or management while also demonstrating innovative and creative practices that have a positive impact on the local community and culture. Such hotels support local arts, participate in corporate social responsibility efforts, and actively implement sustainable development strategies. This includes using environmentally friendly materials and energy-saving measures, gaining recognition within the online industry, and receiving awards for excellence in cultural and creative hospitality. Additionally, examining customer reviews and feedback, especially regarding the hotel's creative elements and social responsibility practices, is essential. Based on the theory of social innovation, theoretical sampling involves synthesizing main themes from deep observation and analysis, presenting significant concepts and patterns. This process evaluates the direct impact factors on social innovation, which include enhancing community welfare, strengthening cultural exchanges, and fostering economic development. Selecting innovative impacts involves practical applications to social innovation, covering improvements in services, enhancing customer experiences, and boosting community involvement in sustainable and long-term innovative strategies.

The case selection is divided into two stages. In the first stage, representative cases in the hotel group industry are collected from journals, newspapers, and the internet, and the basic value strategy unit analysis is established to preliminarily analyze the distribution status of the value activities of creative hotel operators in the sample group. At this stage, the sample selection method is based on the secondary data visibility of cultural creative hotels, and representative shop owners recommended by relevant experts on the website are collected. Finally, significant application results are extracted as samples. Furthermore, based on factors such as the willingness of the supervisor to be interviewed, only two case companies can be selected in the second stage, and further data collection can be conducted through in-depth interviews to examine their value co-creation strategies, core resources, and development processes, as shown in (Table 1).

**Table 1.** Basic information of the case.

| Value Type | Innovative Service Experience Value-Added Type | Original Experience Design Type |
|---|---|---|
| Research Subjects | Taichung The Place Hotel | Chanyee Hotelday |
| Founder | Ching-Po Lin, Wan-Ying Liao | Chun-Lang Dai |
| Established | 2018 | 2016 |
| Capital | USD 320 million | USD 12 million |
| Star Rating | 3-star | 3-star |

| | Features | In a hotel in an art museum, encountering the scenery of humanities and arts in the hotel. | Become a travel destination and build experiences that enhance the quality of the traveler's experience. |
|---|---|---|---|

In addition to secondary sources of information such as journals, reports, books, official websites, and theses, this study chose to interview managers of The Place Hotel and Chanyee Hotelday. These in-depth interviews lasted for more than one hour in each case and were audio-recorded and compiled into a transcript for comparative analysis.

The text analysis herein uses the research method of [39]. It employs highly structured data organization and induction techniques and combines industry data and environmental background knowledge to establish a preset analysis item for the column order interpretation of the collected data. The secondary data sources of this article include news reports, official websites, interviews, and YouTube videos. Through primary data and secondary data collected from multiple sources, as well as company archive data, multiple cross-validation methods are used for triangulation to improve the reliability of the case and ensure data accuracy [38]. Through triangular verification, researchers can comprehensively examine multiple aspects of cultural creative hotels. For example, through in-depth interviews (Appendix A), the emotional stories of employees can be analyzed to understand the core behavioral perspectives of the enterprise. Understanding the performance of corporate culture at various levels permits the interpretation of the actual operation of corporate culture within an organization and analysis of the values behind organizational culture. Yin [34] believes that the triangular validation method can ensure the comprehensiveness of research results, clearly present the relationships between entities in the research domain, and establish the rationality of internal validity.

A total of 7 individuals were interviewed, including 1 actor, 3 collaborative units, and 4 internal unit supervisors (Table 2). The emphasis on the research process strives for the accuracy of data and the objectivity of the researcher, with the rigorous investigation results contributing comprehensively to enhancing the persuasiveness of academic research.

**Table 2.** Profile of respondents and dates

| Code | Job Title | Title | Data Source | Date of Interview |
|---|---|---|---|---|
| A1 | Mr. Jiang | General Manager of The Place Hotel | Face-to-face interview | 4 January 2023 |
| A2 | Dai Junlang | Founder of Chanyee Hotelday | Face-to-face interview | 10 December 2023 |
| A3 | Dai Shuling | General Manager of Chanyee Hotelday | Text mode interview | 9 January 2023 |
| A4 | Ms. Gang | Director of Marketing and Communication | Face-to-face interview | 5 January 2023 |
| A5 | Mr. Zhang | Director of Chanyee Hotelday | Text mode interview | 9 January 2023 |
| A6 | Mr. Douzi | Bowl of Coffee-Owner | Face-to-face interview | 30 January 2023 |
| A7 | Ms. Zhong | Administrative of Art Bank | Face-to-face interview | 9 February 2023 |
| A8 | Mr. Chen | Garden owner of Zhaozhao Tea | Face-to-face interview | 30 January 2023 |

## 4. Case Briefing

### 4.1. Chanyee Hotelday Hualien

Located in eastern Taiwan, Chanyee Hotelday Hualien Mountain Knowledge (hereinafter referred to as Chanyee Hotelday) was established in 2016. The entire hotel is adorned with local indigenous wood and jade, presenting irregular rock-like counters and wave-like ceilings in the lobby. There are 81 rooms in total. Chanyee Hotelday Co., Ltd. (Chiayi, Taiwan) (headquarters) was established in 2008 and is located on Qiming Road in the eastern district of Chiayi City. The organizational structure of Chanyee Hotelday

includes five departments: the Operations Department, Catering Department, Brand Development Department, Resource Integration Department, and Finance Department. Chanyee Hotelday Group is the largest cultural and creative design hotel chain brand hotel in Taiwan.

Chanyee Hotelday and the tourism management strategy are based on the experience of local culture and scenery, creating a brand-new positioning of the experience space of "tourists belonging to other places". Chanyee Hotelday attaches great importance to establishing interactive relationships with customers, actively inviting travelers to participate in brand and marketing activities and providing space for creative ideas. They also invite customers who are interested in the brand to participate in the naming event of the brand mascot "Dream Beast" to establish closer relationships [40]. In addition, Chanyee Hotelday also invites brand customers to participate in an event called "Slow Delivery". This event consists of having customers send themselves postcards, which they receive a year later, creating a warm and moving experience that strengthens the connection with customers. Chanyee Hotelday also provides deeply ingrained and customized services, such as designing local private house attractions and route recommendations for travelers based on the Chanyee Hotelday perspective. The goal is to provide unique and valuable travel experiences and to provide customers with special innovative experience services.

### 4.2. *Taichung Old Master Hotel Group the Place Hotel*

Taichung Old Master Hotel Group The Place Hotel (hereinafter referred to as The Place Hotel) is one of the brand design hotels under the Master Hotel Group. It is located in the Taichung Caowu Road business district within the rich cultural atmosphere in Taichung. Founded in 2018, it has 170 guest rooms and one independent restaurant. The Old Master Hotel Group regards service quality as the top priority and focuses on providing every guest with a warm and comfortable accommodation experience. The hotel attempts to create an environment that feels like home, allowing guests to feel the warmth and convenience of home. During the operating process, it ensures that every detail meets the highest standards. This insistence on quality wins the trust of passengers, as shown by its wins in the Booking.com Traveler Review Awards 2023 and Agoda Traveler Excellence Awards 2023.

In addition, it attaches great importance to team training and development, ensuring that every employee has excellent service quality. Improving the overall service level also encourages employees to be more enthusiastic about their work. This study holds that The Place Hotel brings together local creators and artists, finds surprises and fun through cultural and artistic activities and exhibitions, and provides visitors with different perspectives on Taiwan.

### 5. Findings and Discussion

This study, based on social innovation theory, currently cannot specify which organizations are suitable. Instead, it focuses on the participation of various stakeholders within the social innovation ecosystem to co-create value, continuously advancing organizational value co-creation through interactions and successful social innovations. The evolution of co-creating value within the service ecosystem is a dynamic and interactive process, often triggered by specific situational needs. Organizations build their service ecosystems through strategic alliances, enhancing connections between organizations which effectively promote the realization of social innovation [4].

This study argues that establishing brand positioning and implementing value propositions in cultural creative hotels can stimulate internal cultural awareness and drive cross-industry cooperation in the arts while strengthening the organization's strategic innovation ability. By connecting official and local resources, marketing city tours can be expanded, and cultural creative hotels can be promoted in international tourism exposure.

On the company's core value proposition, the founder of Chanyee Hotelday said Every hotel is a miniature museum, with aesthetics as its core value, connecting with the

local area, and finally being interesting. Adhering to these three principles, high-end but approachable, the more local the more international, displaying the attitude of this city. Hotels should not only be places for dining and sleeping, but also for soul travel (A2). This is the company's mission. "The management team hopes to make our hotel stay an enjoy able part of guests' travel experience in its own right" (A3).

The core values of The Place Hotel enterprise advocate valuing the content of local culture, allowing culture to infiltrate daily life and interact and exchange with an open attitude. Providing various high-quality innovative experiences will bring about changes in value thinking, provided that the richness of life is increased (A1).

*5.1. Social Innovation and Value Co-Creation of Cultural Creative Hotels*

This article will analyze these hotels from the perspective of social innovation ecosystem theory. Cultural creative hotels participate in collaboration through three aspects: the service platform, value co-creation, and the service ecosystem and then add three key elements: resource integration, resource flow and density, and an A2A network to carry out the transformation process of the value activities. The categories of the analysis are as follows, in order: the comparison of performance between social innovation and three-dimensional applications, comparison of activity transformation between social innovation and three key elements, comparison of the social innovation ecosystem and the value co-creation strategy, comparison of connotation and characteristics between the social innovation ecosystem and value co-creation, and analysis of key drivers between the social innovation ecosystem and value co-creation.

5.1.1. A Comparison of the Three Dimensions of Social Innovation

After participating in the three dimensions of social innovation value activities, Chanyee Hotelday optimized cultural aesthetics to create a unique travel experience, collaborated with partners in the application of digital technology to exert innovative influence, and finally established consensus to generate innovative benefits. The analysis is given below and is summarized in Table 3.

(1) Service platform: By providing a structured and feature-rich service platform, optimize and expand diversified partnerships to accelerate the generation of new ideas.
(2) Service ecosystem: In this open innovation culture, the service ecosystem encourages all participants to actively contribute their knowledge, skills, and resources. Provide participants with a space to experiment with new ideas, promote the generation of innovative solutions, and enhance overall system adaptability.
(3) Value co-creation: By creating unique and interactive experiences, it stimulates the enthusiasm and sense of mission of participants in the value co-creation process. Create a cultural emotional connection, establish loyalty and satisfaction among participants, and further drive innovation and value added.

Combining information and communication technology with interactive design can provide unique opportunities for companies to make IT digital technology more user-friendly and develop products and services that can resonate with emotions. This helps to "deepen the image of cultural and creative design and creates a workplace that employees are eager to join (A5)".

After participating in the three-dimensional value activities of the social innovation foundation, The Place Hotel has demonstrated an innovative experience in art services, thereby deepening the image of the enterprise, highlighting the characteristics of art hotels in the market, stimulating collaborative innovation issues, and expanding value co-creation benefits. The analysis is given below and is summarized in Table 3.

(1) Service Platform: Optimizing diverse partnerships and revitalizing the network of innovative value activities. The service platform enhances interactions with various partners, increasing the momentum and visibility of innovative activities.

(2) Service Ecosystem: Cultivating an open innovation culture and encouraging contributions from all participants. This strategy aims to create an environment that supports continuous innovation and collective growth, offering all participants opportunities to leverage their strengths.

(3) Value Co-Creation: Creating unique experiences that inspire a sense of mission in value co-creation among participants. By providing unique cultural experiences, this approach stimulates the creative potential of participants, collectively shaping and expanding new value.

The Place Hotel attaches great importance to selecting partners, actively promoting cultural and creative imagery, and combining art with local cultural resources to provide distinctive and innovative service experiences.

Efforts are made to deeply integrate local culture and art, demonstrate brand positioning, and emphasize that "the National Art Museum and the Art Bank are absolutely indispensable partners" (A1).

**Table 3.** A comparison of the three aspects of social innovation in cultural creative hotels

| Three Facets of Social Innovation | Service Platform | Service Ecosystem | Value Co-Creation |
|---|---|---|---|
| Chanyee Hotelday | Expand the platform network, network engagement. | Promote the scope of social responsibility for sustainability. | Interaction amplifies value. Transmit energy and create valence. The extension of value co-creation. |
| The Place Hotel | Optimize the network of innovative value activities based on multiple partnerships. | Foster a culture of open innovation and encourage all parties to participate and contribute. | Create unique experiences to trigger. Participant value co-creation mission. |

In sum, after comparing the three dimensions of social innovation in The Place Hotel and Chanyee Hotelday, this study found that there were slight differences in the two cases across the three dimensions of social innovation. Chanyee Hotelday focuses on expanding the interaction and visibility of participants with service platforms while promoting the scope of social responsibility and expanding the communication energy of value transmission. The Place Hotel focuses more on optimizing partnerships, cultivating open and innovative cultural services, and creating unique experiences, triggering a sense of mission for participants to jointly create value.

5.1.2. A Comparison of the Transformation of Three Value Activities in Social Innovation

After expanding three key value activities through social innovation, Chanyee Hotelday has rapidly developed digital technology and integrated resources, promoted communication and interaction networks, and formed the characteristics of an art and cultural creative hotel, leveraging the mutual benefit and shared value of partners. This study's analysis is given below and is summarized in Table 4.

(1) Resource integration: This strategy promotes the active participation of partners and also encourages them to contribute their unique innovative thinking and professional skills.

(2) Resource flow and density: The key to establishing mutual trust lies in transparent communication and shared values, which can create a supportive cultural environment that includes good organizational governance and ethical practices.

(3) A2A network: By establishing an (Artist-to-Artist) network, cultural entrepreneurs can learn from each other, share market experience and creative ideas, and thereby improve the overall market competitiveness of creative products.

Chanyee Hotelday has never taken price orientation as its business philosophy. Chanyee Hotelday places more emphasis on exploring how to tell exciting stories for this land through the interface of hotels and travel, so that "more people can discover the beauty of Taiwan" (A3).

(1) Resource Integration: By integrating resources from multiple sources, an interactive network has been created that resonates emotionally among participants. This interaction not only enhances the depth and breadth of cultural exchange but also strengthens the connection between participants.

(2) Resource Flow and Density: Facilitating the effective flow of resources and increasing interaction density establishes a strong consensus and optimizes the participation of partners. This enhancement in resource mobility accelerates the innovation process across the network, fostering active value creation during the co-creation process.

(3) A2A Network: The expansion of communication and cooperation between alliances through the A2A (Artist-to-Artist) network establishes closer connections. This strategy not only enhances brand charisma but also creates uniqueness and fosters innovation in the cultural and creative sectors.

**Table 4.** A comparison of the transformation of the three major value activities of cultural creative hotels and social innovation.

| Social Innovation Three Important Items | Resource Integration | Resource Flow and Density | A2A Network |
|---|---|---|---|
| Chanyee Hotelday | Engage partners and contribute innovative thinking and capabilities. | Initiate a relationship of mutual trust and promote the development of local art and culture. | Integrate cultural entrepreneurs to expand the visibility of cultural and creative products. |
| The Place Hotel | Expand local cultural and artistic exchanges and cooperation, establish a local cultural brand image, and show the uniqueness of art, culture, and creativity. | Emphasize the rapid flow of resources and provide a diverse service experience. Engage partners and contribute innovative thinking and capabilities. | Establish a people-oriented interactive communication network, and the interactive network transmits value and arouses emotional resonance. |

Research has found that analyzing and strengthening the interaction between service ecosystems can expand the scope of sustainable social responsibility, which is consistent with Chen's [3] findings on innovative services for sustainable development in the hotel industry. As one informant said, "Local leaders have long advocated for a hotel in Hualien that would allow visitors to experience the local customs" (A5). This not only helps to enhance the competitiveness of cultural creative hotels but also emphasizes the overall impact of local cultural and artistic development on society and the environment.

This study analyzes the transformation of value co-creation activities in the social innovation ecosystem based on the basic theories of service platforms, service ecosystems, and value co-creation, as well as the interactive process between them. In the social innovation ecosystem, the A2A (Actor-to-Actor) network promotes mutual understanding and collaboration among stakeholders. This network can promote value co-creation by establishing mutual trust, sharing resources, and jointly solving problems. Resource flow is a key element in the social innovation ecosystem, and effective resource flow helps to achieve resource complementarity and sharing, promoting innovation and value creation. The impact of resource density on cooperation means that more resources are available to

participants, which helps to improve the opportunities and benefits of cooperation. The conceptual framework of the proposed social innovation ecosystem is shown in Figure 1.

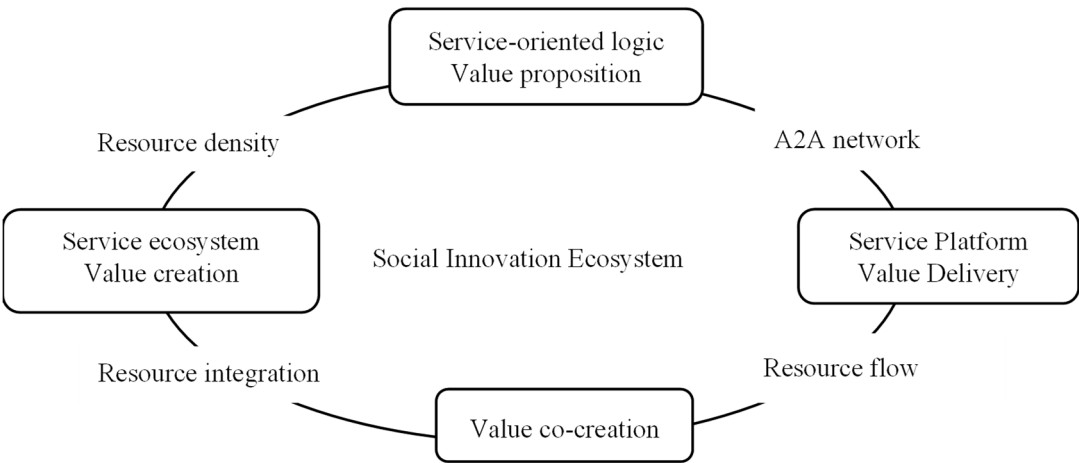

**Figure 1**. The conceptual framework of a social innovation ecosystem.

*5.2. Analyzing the Connotation Characteristics of Social Innovation and Value Co-Creation in Cultural Creative Hotels*

5.2.1. Social Innovation Ecosystem and Value Co-Creation

When exploring the organization of the social innovation ecosystem, how to combine the participation and collaboration of various stakeholders becomes a key question. What are the elements of promoting a value co-creation strategy? The achievement of value co-creation through important key strategies will be compared with the analysis of the social innovation system concept framework to promote value co-creation. The following analysis is organized as shown in Table 5.

**Table 5.** Value co-creation operation strategies based on the three dimensions of social innovation analysis.

| Value co-creation operation strategy | The co-evolution of the service ecosystem. | The construction of the service platform was started. | Establish synergy and participate in value co-creation. |
|---|---|---|---|
| Chanyee Hotelday | Increase participant interdependence and create a friendly cycle of reciprocity. | Actively participate in the needs of participants, and obtain feedback based on them. | Generate identity and mutual trust and attract more working partners to coexist and co-create. |
| The Place Hotel | Deeply understand the needs of participants and strengthen recognition and support. | Make use of local resources to stimulate the potential and contribution of participants. | Form a mutually beneficial and symbiotic relationship, create value, and create continuity. |

In the social innovation ecosystem of cultural creative hotels, enterprise organizations establish clear value propositions and construct the following three value co-creation operation strategies based on the three dimensions of social innovation: (1) the co-evolution of the service ecosystem; (2) the initiation of service platform construction; (3) and the establishment of collaborative participation in value co-creation for value activities.

After participating in the social innovation ecosystem value co-creation strategy, Chanyee Hotelday plays a key role in guiding stakeholders both inside and outside the

organization to participate in value creation interactions, while online interactions enhance the flow of value transmission. This study finds that the following occurred:

(1) The Co-evolution of the Service Ecosystem: This involves forming interdependent relationships with other organizations and individuals, actively and continuously establishing interactive relationships. During the evolutionary process, participants create a tighter and more flexible network to collectively address external challenges.

(2) The construction of the service platform was started: This involves actively responding to the needs of participants and garnering valuable feedback to optimize services. The platform not only provides a space for customer interaction but also serves as a center for innovation and creative exchange, enhancing the connection between the organization and its customers.

(3) Establishing Collaborative Participation in Value Co-creation: Through the co-created service platform, partners are able to engage in deep cooperation based on mutual trust. This not only attracts more partners but also fosters a symbiotic and mutually beneficial relationship, enhancing the innovative experience for guests.

Chanyee Hotelday takes becoming a travel destination as its core concept and constructs all experiences that can enhance the sensitivity of travelers (A2).

Through participating in the social innovation ecosystem value co-creation strategy, The Place Hotel promotes more active participation and consensus among stakeholders in innovation activities, forms a bridge for value creation, enhances the recognition of value goals, and makes them more willing to participate in the co-creation process. This includes the following:

Inviting grassroots employees from franchise store members to participate in the discussion of the plan, adjusting the reception pipeline, establishing a carefully designed framework model, and introducing the interaction between people when hotel guests come directly to the store. We have adjusted our system and scale planning methods to quickly improve our service satisfaction and reputation. It also includes collaborating with various departments and providing frontline services at any time, resulting in a rapid increase in product satisfaction (A1).

(1) The Co-evolution of the Service Ecosystem: This involves a deep understanding of participants' needs, optimizing recognition and support for participants, enhancing the customer service experience, and deepening the connections between employees and partners. This fosters a stable ecosystem that promotes long-term collaboration and innovation.

(2) The construction of the service platform was started: This utilizes local resources to stimulate the potential and contributions of participants, increasing the efficiency of innovation activities. This promotes the establishment of close ties with the local community, responds to market demands, and quickly adjusts service plans, thereby enhancing customer satisfaction and brand value.

(3) Establishing Collaborative Participation in Value Co-Creation: This forms mutually beneficial symbiotic relationships through continuous value co-creation activities, solidifying these relationships while continuously attracting new partners and customers. This promotes the development of an internal culture of innovation and creativity within the organization.

Internal members receive tailored professional training and introduce a service design thinking model. This enables internal management integration to constantly change roles and participate in innovative service experiences, while department members establish interactive networks through professional training and learning, gather emotions face-to-face with partners, and add value to enhance service experiences (A4).

This study found that after comparing and analyzing the three dimensions of value activities and value co-creation strategies, differences in specific practical methods and strategies between the two cases remain. Chanyee Hotelday focuses on content development and unique travel experiences. Original experience designs can attract customers

seeking unique experiences. By contrast, The Place Hotel focuses on resource integration and innovative service exchange models, while the added value of innovative service experience is suitable for customers who pursue surprise and diversity. This study suggests that the construction of appropriate value co-creation strategies based on the advantages of the enterprise organization and customer goals depends on the positioning of the target market.

5.2.2. A Comparison of the Connotation and Characteristics of Value Co-Creation in Innovative Social Ecosystems

The social innovation ecosystem is a dynamic and interrelated system, in which various components interact and continuously develop through value transmission. Vargo et al. [25] contended that the sustainability elements required to serve ecosystems are often overlooked, but these elements are important for achieving institutional innovation. In addition, Chen [41] noted that sustainability elements also include organizational culture and values, which are crucial for encouraging and accepting innovative thinking. Sustainability elements are not only a short-term event for achieving innovation sustainability but also a process that requires continuous investment and long-term support. Value proposition can be seen as regulating the value mechanism, balancing the consistency of value propositions among different stakeholders, and enabling enterprises to continue to move forward in value co-creation [9]. Based on this, the hidden strategic connotations and characteristics in the social innovation ecosystem enrich the value co-creation and cultivation of cultural creative hotels, one of the reasons for maintaining the continuous operation of the organizations.

Therefore, based on the premise that enterprise organizations adhere to sustainable development within the market environment, the social innovation ecosystem provides a sustainable innovation field environment. This article summarizes the connotation and characteristics of Chanyee Hotelday's value co-creation in the innovative social ecosystem. This is summarized in Table 6.

(1) Value co-creation connotation:
  (a) Sensory Experience of Intellectual Services: Through ICT interactive design systems and proactive app linkage notification systems, an interactive and personalized customer innovation experience is provided, enabling customers to receive real-time updates on services.
  (b) Experience in 3D Virtual Reality Modules: Three-dimensional virtual reality technology allows customers to experience the hotel's environment and local cultural activities. Technological innovations expand the range of customer experiences and participation.
  (c) Innovative Integration of Indigenous Culture and Artist Collaboration: Engaging in collaborations with Indigenous communities and artists, blending traditional cultural elements with modern artistic creativity, and creating unique cultural experience activities. The service's cultural depth supports the preservation and innovation of local culture.
(2) Value co-creation characteristics include the following:
  (a) Local aesthetics and cultural creativity trajectory: Focuses on integrating local aesthetics and developing humanistic creativity, establishing the brand's uniqueness and depth while attracting travelers seeking non-traditional travel experiences.
  (b) ICT interactive housing design: Combining the latest technology such as smart home guest room services, housing design not only enhances personalized innovative services but also strengthens interactions between customers and cultural creative hotels.
  (c) Exploring Indigenous culture to enrich innovative elements: Through the rich elements of Indigenous culture, cultural heritage is combined with modern

travel demands to develop new market opportunities while also contributing to the promotion of Indigenous cultural crafts.

The connotation and characteristics of The Place Hotel's value co-creation in innovative social ecosystems will be analyzed below.

(1) Value co-creation connotation:

(a) Applying art museum art and local cultural innovation experience: The hotel integrates art museum exhibits and local culture to provide guests with a rich humanistic experience. This aesthetic appeal serves as a conduit connecting visitors with local cultural innovations.

(b) By leveraging information technology and digital economic marketing strategies, the hotel fosters interdisciplinary collaborations that bring aesthetic life experiences into daily interactions, thereby stimulating customer interest and attracting a diverse clientele.

(c) The management, driven by knowledge and diversity, utilizes a wide range of expertise to create captivating activities that attract guests. This management style is crucial for developing marketing strategies that resonate across various guest preferences and cultural backgrounds.

(2) Value co-creation characteristics include the following:

(a) Collaborative partners enhance brand image: Brand image through collaboration: The collaborative efforts between the hotel and its partners optimize the brand image, demonstrating a mutual commitment to marketing and customer engagement that benefits all parties.

(b) Breakthrough innovative design co-creation cooperation: Co-creation challenges the traditional boundaries of hotel service design, introducing groundbreaking concepts that transform guest experiences. This collective creativity leads to the development of novel services that redefine the hospitality industry.

(c) Establishing alliance co-creation activity strategies: The hotel formulates strategies for alliance-driven co-created experiential travel activities. These strategies aim to build and guide cooperative efforts towards shared objectives, enriching the travel experience through curated cultural and recreational outings that highlight the destination's uniqueness.

**Table 6.** A comparison of the connotation and characteristics of value co-creation in social innovation ecosystems.

| Cases | Value Co-Creation Connotation | Value Co-Creation Characteristics |
|---|---|---|
| The Place Hotel | Apply art museum art and local people's cultural innovation experience. Use IT digital economy marketing strategies. Provide knowledge-based multi-collaborative management. | Collaborate with partners to enhance brand image. Co-create collaborations with breakthrough innovative designs. Establish an alliance co-creation activity strategy. |
| Innovative types of value-added service experiences | | |
| Chanyee Hotelday | Intellectual service for the five senses experience. Add APP active link notification system. Experience the 3D virtual reality module. Innovation combines Aboriginal culture and artist collaboration. | Local aesthetics, humanistic and creative trajectories. ICT interactive system housing design. Exploring Indigenous culture and enriching innovative elements. |
| Original Experience Design Type | | |

They certainly respect us, so this is a joint effort of the community. They showcased us on their official website, and my focus was particularly on tea. This resonates, especially

for unique or specialty stores. This is the right approach, and we are very willing to cooperate with them in this regard (A8).

Chanyee Hotelday is proud of its art rooms, and the design director points out that well-known artists or creators are invited to participate in the creation of the rooms. For example, Hualien's local products are used to develop a unique design style for each room, showcasing the diverse aspects of Hualien through stone, wood, and paintings. Among them, "The Mountain" was designed by the Indigenous Amis wood sculptor Iyau Kajo, "The Moment" was designed by a visual artist using composite media, and "The God of the Mountain is Coming" was created by an illustrator. Chanyee Hotelday also launched an exclusive partnership with Gekkei Rent-A-Car. All partners are invited to participate in the synergy of creation and co-prosperity.

In sum, this study found that from the perspective of service ecosystem theory, value co-creation is the result of creative interaction and exchange, which are complementary. This further extends the view that in cultural creative hotels, stakeholders cultivate the ability to integrate resources between collaborations and expand their communication between local culture and art, leading to spillover effects.

Babu et al. [4] argue that strategic alliances constitute value co-creation in the service ecosystem. The value cultivation of cultural creative hotels carries the advantages of local culture and characteristics, promoting close interaction among cultural creative hotel alliance partners (such as artists, cultural entrepreneurs, and other industry stakeholders), enriching the local cultural and artistic atmosphere, jointly stimulating cultural consensus among partners, strengthening innovative services, and creating a virtuous cycle of value co-creation.

*5.3. Explore the Key Elements Driving the Social Innovation Ecosystem of Cultural Creative Hotels*

Through complex networks and co-evolutionary processes, the social innovation ecosystem will transmit value in the best form through the value of innovative services.

Analysis of Social Innovation Ecosystem and Key Trends

Under the concept of value co-creation, it is possible to lead stakeholders from various departments of the organization to achieve co-creation goals through value consensus and mutual benefit. At the same time, who drives the process of value creation? How do all parties interact and exchange ideas together? Why effectively convey social innovation concepts to stakeholders and obtain their positive response and participation? Under the conceptual framework of the social innovation ecosystem, organizing innovation network interactions based on value co-creation activities promotes co-evolution, initiates construction, and establishes an operational mechanism for partner participation. In the process of the co-evolution of cultural creative hotels, value creation stimulates consensus and cooperation among participants through effective cognitive and value exchange activities. When constructing a sound system to initiate social innovation activities, actors must further optimize the connections between different participants in the ecosystem. Finally, the organization must establish a collaborative participation relationship to form a dynamic social innovation ecosystem network structure.

Through a series of social innovation cycles, innovative thinking and transformation will emerge. Therefore, three key driving factors of the social innovation ecosystem are inferred: (1) the communication of cognition and values [42]; (2) reaching consensus through role shifting among participants [43]; and (3) interdependence and identification between organizations [44]. This study constructs the conceptual framework of the expanded social innovation ecosystem as shown in Figure 2.

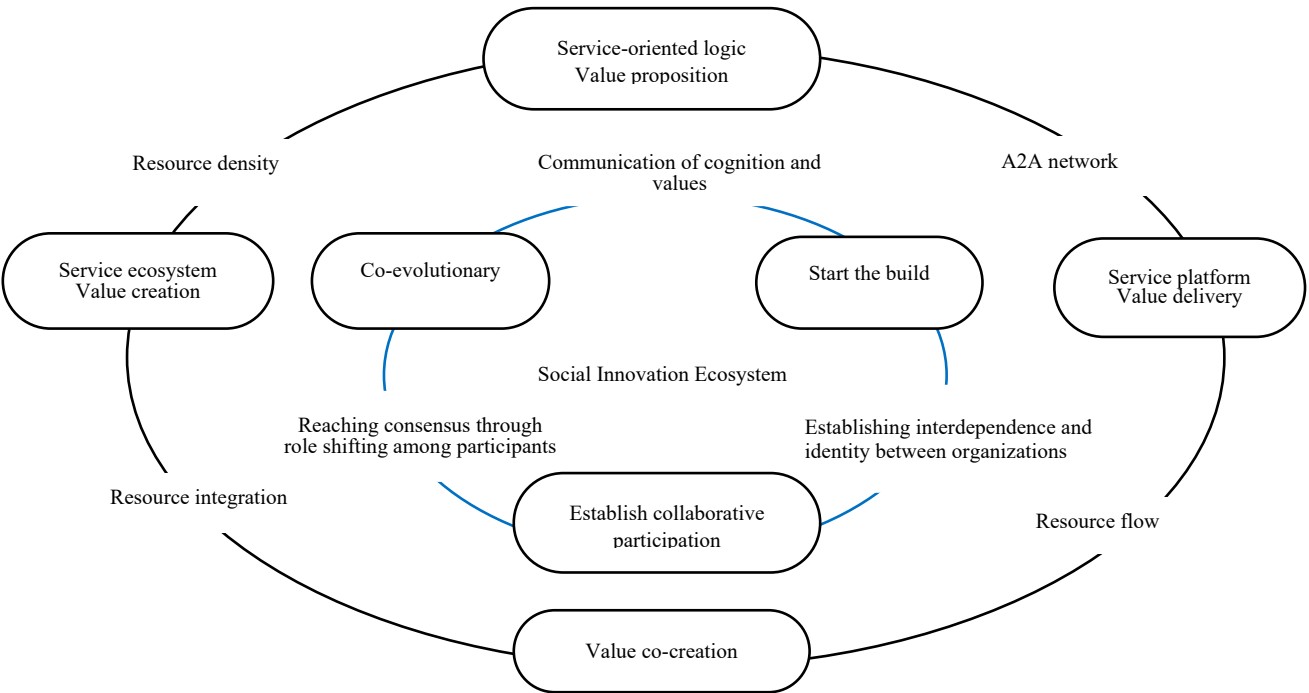

**Figure 2.** The conceptual framework of an expanded social innovation ecosystem. Source: organized by this study.

After the emergence of key drivers in the social innovation ecosystem, a series of value transmissions are initiated simultaneously, and then through the transformation-driven strategy mechanism of communication and interaction, consensus and support are generated, triggering innovation performance and promoting value actions of benefit sharing.

(1)  The key drivers of the innovation ecosystem of Chanyee Hotelday cultural and tourism society are presented below.

    (a)  Communication of cognition and values: This facilitates the exchange of identification and thought processes, successfully optimizing the environment for engagement. Creating a comprehensive environment encourages employees and partners to share insights and values, thereby enhancing team cohesion.

    (b)  Reaching consensus through role shifting among participants: Initiating changes in roles and stances not only helps in forming consensus and trust but also fosters deeper collaboration and understanding. This transformation enables participants to adapt organizational strategies in response to market changes during the innovation process.

    (c)  Establishing interdependence and identification between organizations: Through mutual dependence and identification among organizations, a network that supports energy cycles is established, allowing each department to receive support from others. This interdependence enhances organizational vitality, improves the efficiency of internal management, and boosts overall innovative execution capabilities.

(2)  The key drivers of the social innovation ecosystem of The Place Hotel are as follows:

    (a)  Communication of cognition and values: By fostering the recognition of values associated with social innovation, we encourage both internal and external participants to contribute jointly. Interactive exchanges enhance consensus between teams and partners.

    (b)  Reaching consensus through role shifting among participants: Transforming the roles and relationships of participants enhances identification through increased

interaction modes, simultaneously strengthening the organization's capacity to adapt to environmental changes.

(c) Establishing interdependence and identification between organizations: Continuously enhancing our own innovative capabilities, optimizing the interdependence within the organization, and establishing a mutually supportive network, thereby creating value between organizations and individuals.

The Place Hotel has shown a relatively open attitude in its operations. Although it is necessary to consider the limitations of corporate image and passenger needs when promoting certain works, in the hotel industry, The Place Hotel is more willing to discuss the content of cooperation with partners and demonstrate marketing techniques with regenerative capabilities (A7).

According to [3], incorporating local art and culture into service design can continuously provide customers with unique value and experience. This innovative service experience model has a profound impact on the sustainable operation policy of enterprises. In addition to reshaping the innovative thinking of enterprises, it also stimulates breakthrough innovation capabilities. We present our analysis of the key drivers of value co-creation in the social innovation ecosystem in Table 7.

**Table 7**. An analysis of the key drivers of the social innovation ecosystem.

| Cases | Chanyee Hotelday | The Place Hotel |
|---|---|---|
| Key drivers | Original experience design-type innovation. | Value-added service experience. |
| Communication of cognition and values | Promote the exchange of ideas and identity and create momentum for active participation. | Promote the value recognition of social innovation and be willing to contribute collaboratively. |
| Reaching consensus through role shifting among participants. | Initiating a role reversal stance to generate a good relationship of consensus and trust. | Participate in role changes at any time and motivate the motivation of the interaction mode. |
| Establish inter-organizational interdependence and identity | Creating identity drives the vitality of departments and organizations. | Continuously improve their own innovation ability and give full play to their value functions. |

At the same time, The Place Hotel collaborates on a practical shared value model to create a unique traditional travel experience, allowing travelers to enjoy rich and interesting life experiences while exploring urban elements.

As one participant stated, All members are willing to continue working together and participate in the next stage of strategic marketing plans, which proves the success of the entire social innovation ecosystem (A6).

Our research results show that in order to maintain the elements of sustainable value co-creation, the key focus is on the same goal direction of constructing key drivers for value co-creation, which also means that key drivers will affect the sustainability of the social innovation ecosystem.

Meiser et al. [31] and other scholars propose that the executors of social innovation strategies should not only lead the strategy but also strengthen the application of co-creation in the process of social innovation. This study suggests that the co-creation of application prerequisites must be achieved through the transformation of roles and cognitive communication among actors or stakeholders in the ecosystem, in order to generate mutual value co-creation thinking and promote the subsequent development of the social innovation ecosystem through the evolutionary process. From the perspective of value co-creation theory, the integration and collaboration of resources between enterprise organizations and stakeholders is achieved through creating and providing value to achieve common interests [8].

In sum, how to maximize the value and benefits of the deep structure of the social innovation ecosystem is a highly challenging issue. Therefore, the social innovation ecosystem with the goal of value co-creation will ultimately play a role in establishing consensus and interdependence through a series of processes such as exploration, communication and dialogue, interaction, cognition, empathy, and role exchange, in order to drive the innovation of the social innovation ecosystem for sustainable development. At the same time, the three key forces of value co-creation in the social innovation ecosystem will emerge.

## 6. Conclusions and Suggestions

### 6.1. Conclusions

This study explores the process of value co-creation in the social innovation ecosystem of cultural creative hotels, gradually forming a dynamic network through participating in evolution and constructing collaborative interactions.

First, based on the theoretical foundation of social innovation strategy alliances and value co-creation, the viewpoints of previous scholars are extended and applied to the innovation ecosystem environment of cultural creative hotels, hoping to drive local mutual benefit and prosperity through promoting value co-creation. Second, this study extends the view of [31] and other scholars that social innovation must be based on co-creation, and the social innovation strategy alliance and value co-creation viewpoint identified by [3] are applicable to the social innovation ecosystem of cultural creative hotels. It also reinforces the scope of application proposed by the following.

(1) Inter-organizational interdependence and identity establishment: Collaboratively establish a set of clear goals and visions with partners, which should reflect the mutual interests of the organizations involved and address their key issues. (2) Enhancing exchanges of perceptions and values: Cultural and creative hotels organize workshops and training events to deepen the understanding and consensus of internal organizational cultural values while also establishing a common value perception among participants to foster a sense of identification with each other. (3) Transformation of roles among participants: Establish internal training workshops that encourage staff and partners to shift between different role functions, enhancing the team's flexibility and multifunctionality. (4) Through a value co-creation strategy alliance for social innovation, cultural and creative hotels strengthen the close cooperative relationships among ecosystem organizations, facilitating the exchange of value perceptions, thereby enhancing the efficiency and vitality of the entire social innovation ecosystem and increasing the practical value to further leverage the crucial mechanisms of social innovation [3].

Collaborating with local artists or business groups jointly promotes the preservation and innovation of local culture. Expanding cross-organizational strategic meetings allows external stakeholders of cultural and creative hotels to share resources, enhancing value creation and strategic alliances [3]. This collaboration builds trust and transparency and fosters interdependence among parties. Thus, when stakeholders engage in social innovation activities collaboratively, they extend the scope of value creation [22]. Establishing external communication platforms and periodically promoting exchange collaborations strengthens the service platform's interaction network [26], linking local creative cultural events and encouraging community and stakeholder participation in resource sharing to drive social innovation. Cultural and creative hotels establish cross-functional teams to address specific issues, enhancing team members' experiences in different roles, improving understanding, stimulating innovative thinking, and advancing the organization's network relationships to optimize service innovation capabilities [8].

Finally, this study suggests that after the exchange of cognition and values between cultural creative hotel organizations, consensus among participants can be generated, and through the transformation of roles among participant relationships, the benefits of value co-creation can be enhanced. Therefore, after comparing the value co-creation strategies

of cultural creative hotels, it was found that after the actors initiate various institutional and social innovation activities, they will motivate the various departments to continuously create a driving force for participation and cooperation.

In sum, the main contribution of this article is to explore the three key elements of the social innovation ecosystem driven by cultural creative hotels: (1) establishing interdependence and identification between organizations; (2) strengthening cognitive and value exchange between each other; and (3) establishing consensus through role switching among participants. This ultimately attracts partners to participate in value co-creation, becoming a new guide for the operation of value co-creation models in innovative social ecosystems. The insight of this study is that through complex networks and co-evolutionary processes, the social innovation ecosystem will transmit value in the best form through the value of innovative services.

### 6.2. Research Limitations and Follow-Up Recommendations

This study analyzes the value co-creation process of Chanyee Hotelday and The Place Hotel as a pioneering case and explains the significance of value co-creation in the actual context of social innovation ecosystem. Pinto [45] emphasizes that the social benefits of social innovation are uncertain, and the important point is that different stakeholders may have different perspectives and expectations, and social innovation may vary depending on the perspectives of participants.

The above situation may impose several limitations on this study: (1) Because of the large scale of most hotel groups, it remains difficult to determine whether the two case chain hotels in this study are small-scale cultural creative hotels and whether their short operating experience is a successful case or a sufficiently representative case. This may cause bias in the selection of research subjects. (2) Due to the relatively limited amount of secondary data and related research in this industry ecosystem, this study may rely more on the subjective statements of respondents in constructing case data. Although the researchers have made efforts to search for multiple data sources to improve the reliability of this study's data, there may still be omissions. This study found that there may be challenges over time. Thus, it is recommended that the number of similar studies in the future should be increased.

**Author Contributions:** Conceptualization, C.-L.C.; Methodology, C.-L.C.; Validation, M.-R.W.; Formal analysis, M.-R.W.; Investigation, M.-R.W.; Writing—original draft, M.-R.W.; Writing—review & editing, C.-L.C.; Visualization, C.-L.C.; Supervision, C.-L.C. All authors have read and agreed to the published version of the manuscript.

**Funding:** This research was partly funded by National Science and Technology Council, Taiwan. (Grant No. MOST 111-2410-H-144 -005 -MY2).

**Institutional Review Board Statement:** Not applicable.

**Informed Consent Statement:** Not applicable.

**Data Availability Statement:** The data presented in this study are available on request from the corresponding author.

**Conflicts of Interest:** The authors declare no conflict of interest.

### Appendix A. Interview Guide

| | |
|---|---|
| 1. | Please briefly describe the process of establishing a cultural creative hotel? |
| 2. | How has the business performance of cultural creative hotels been in recent years? |
| 3. | How is the pricing for cultural creative hotels? Will you refer to the pricing of other cultural and tourism options? |
| 4. | Can you talk about the recent upgrade and application process of online booking platforms? |
| 5. | In terms of personnel allocation in hotel management, cultural creative hotels are mainly divided into room service departments, catering staff and counters, procurement departments, and marketing departments. How do we conduct human resources training in internal management? |

| | |
|---|---|
| 6. | Please explain how to import the local cultural and time imprint experience service ecosystem, and why did you import the backpack travel service for travelers in the first place? How to persuade and educate employees and consumers to use this innovative marketing strategy? In addition, what are the differences between cultural creative hotels before and after importing the system? |
| 7. | Please explain the operation of the cultural creative hotel, as well as the interaction with employees. In addition, how were the related equipment and exhibition halls in the museum initiated and co-created? |
| 8. | Please explain the process of establishing a cultural intellectual journey. How to collaborate with other industries and communicate with internal employees, share resources and cultural construction knowledge, and establish mutual benefit? And what kind of results does the verification of local cultural intellectual journey bring to cultural creative hotels? |
| 9. | May I ask what other important partnerships are there in the overall upstream and downstream operations of cultural creative hotels, besides hotel groups? Why are they important to cultural creative hotels? |
| 10. | Finally, in the process of introducing local culture into cultural creative hotels, which link has the greatest impact on the transformation of cultural creative hotels? |

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
