# Peer review of "Exploring the Value Co-Creation of Cultural Creative Hotels: From the Perspective of Social Innovation"

_sustainability, doi:10.3390/su16114510_

Round 1

Reviewer 1 Report

Comments and Suggestions for Authors

Review article titled “Exploring the value co-creation of cultural creative hotels: From the perspective of social innovation”, Sustainability.

1.      The study discusses an innovative and important topic, value co-creation in cultural creative hotels.

2.      The attempt to create research value by the authors is recognized. Thus, the comments that follow aim to help improve the paper.

3.      The paper seems to need a radical restructuring and more clarity.

4.      The abstract requires to be written more clearly.

5.      “Cultural creative hotels” need to be clearly defined the first time they are referred.

6.      “Social innovation” needs to be clearly defined the first time they are referred.

7.      Some cites are repeated (e.g., line 51).

8.      The methodology needs to be written clearly  and concisely.

9.      The paper needs clear reference to what applies within a given organization and what applies to interactions among organizations concerning different aspects discussed in the paper (e.g., social innovation).

10.  In diverse instances, the paper repeats the same ideas (e.g., what is written in the main body and what is shown in the tables).

11.  General ideas from the literature dominate the paper without clearly relating them to the empirical data.

12.  The paper is written unclearly. It is, at times, confusing (see as examples lines 204-212, 233-240, and 319-336) and lacks flow.

13.  It may be desirable to incorporate cites from the interviews as well as explicitly use data from the secondary sources. Some interviewees’ information from Table 2 is absent in the paper.

14.  Among others, summaries (e.g., lines 431-437and lines 458-464) and tables (e.g., Table 3) need to provide specifics.

15.  By and large the paper is unclear, over-emphasizes the theoretical without clearly relating it to the empirical, and it lacks rigor and flow.

Comments on the Quality of English Language

The paper needs to be re-written. It lacks flow, and clarity.

Author Response

Dear reviewer,

Thank you for giving us the opportunity to revise and resubmit our manuscript. After carefully reviewing the attached comments, we have revised the manuscript to address the concerns raised by the reviewers. The following contains detailed responses to the suggestions made by the reviewers. We hope that with these revisions, our manuscript meets the quality requirements of your journal.

Reviewer 1:

  1. The study discusses an innovative and important topic, value co-creation in cultural creative hotels.

AR: Thanks for your comment.

  1. The attempt to create research value by the authors is recognized. Thus, the comments that follow aim to help improve the paper.

AR: Thank you for your comprehensive suggestions and guidance. In the following we address the reviewers’ comments one by one.

  1. The paper seems to need a radical restructuring and more clarity.

AR: Thanks for your suggestions. We have revised and restructured this article to make it more clarity.

  1. The abstract requires to be written more clearly.

AR: Thanks for your suggestions. We have rewritten the abstract.

  1. “Cultural creative hotels” need to be clearly defined the first time they are referred.

AR: Thanks for your suggestions. We have added the definition of cultural creative hotels.

“A cultural creative hotel can be defined as a for-profit organization that incorporates local culture and unusual design elements in its service offerings, which include accommodation and experiential activities with cultural and recreational value.”

  1. “Social innovation” needs to be clearly defined the first time they are referred.

AR: Thanks for your suggestions. We have added the definition of social innovation.

  1. Some cites are repeated (e.g., line 51).

AR: Thanks for your suggestions. We have deleted the repeated cites.

  1. The methodology needs to be written clearly and concisely.

AR: Thanks for your suggestions. We have added some text to make the methods clearly and concisely.

  1. The paper needs clear reference to what applies within a given organization and what applies to interactions among organizations concerning different aspects discussed in the paper (e.g., social innovation).

AR: Thanks for your suggestions. We have added some text to show the interactions among organizations concerning different aspects discussed in the paper.

  1. In diverse instances, the paper repeats the same ideas (e.g., what is written in the main body and what is shown in the tables).

AR: Thanks for your suggestions. We have re-organized the text and make it more concise.

  1. General ideas from the literature dominate the paper without clearly relating them to the empirical data.

AR: Thanks for your suggestions. In this revision, we have re-organized the text and make the literature and discussions relating to the empirical data.

  1. The paper is written unclearly. It is, at times, confusing (see as examples lines 204-212, 233-240, and 319-336) and lacks flow.

AR: Thanks for your suggestions. According the reviewer’s guidance, we have revised the content.

  1. It may be desirable to incorporate cites from the interviews as well as explicitly use data from the secondary sources. Some interviewees’ information from Table 2 is absent in the paper.

AR: Thanks for your suggestions. We have incorporated the interviews data, for example the informant A3, A5 etc.

  1. Among others, summaries (e.g., lines 431-437and lines 458-464) and tables (e.g., Table 3) need to provide specifics.

AR: Thanks for your suggestions. We have provided specifics and make them clear, for example in 5.1.1.

  1. By and large the paper is unclear, over-emphasizes the theoretical without clearly relating it to the empirical, and it lacks rigor and flow.

AR: Thanks for your suggestions. In this revision, we have added text to link the rationale between data and findings.

Reviewer 2 Report

Comments and Suggestions for Authors

Overall, the article provides a valuable exploration of the social innovation and value co-creation within the context of cultural creative hotels. However, there are several areas that could benefit from further elaboration, clarification, or improvement:
1) While the article mentions adopting a qualitative research approach, it would be helpful to provide more details on the specific methods used for data collection and analysis. How were cultural creative hotels selected for study? What criteria were used to identify key factors driving social innovation? 
2)  The article discusses the process of value co-creation in the social innovation ecosystem of cultural creative hotels but could benefit from clearer definitions and conceptualization of terms such as “social innovation,” “value co-creation,” and “innovation ecosystem.” Clarifying these concepts would help readers better understand the theoretical framework guiding the study.

3) The article proposes practical recommendations for corporate organizations in the field of social innovation, but it could provide more specific and actionable suggestions. How can cultural creative hotels effectively implement the three key elements of the social innovation ecosystem identified in the study? Offering practical guidance would increase the relevance and applicability of the research findings.

4) The authors need to engage in more discussion with previous literature in the conclusion section to highlight the theoretical value of this study. Overall, I feel that the theoretical value of this article is low.

Comments on the Quality of English Language

no

Author Response

Dear reviewer,

Thank you for giving us the opportunity to revise and resubmit our manuscript. After carefully reviewing the attached comments, we have revised the manuscript to address the concerns raised by the reviewers. The following contains detailed responses to the suggestions made by the reviewers. We hope that with these revisions, our manuscript meets the quality requirements of your journal.

Reviewer 2:

Overall, the article provides a valuable exploration of the social innovation and value co-creation within the context of cultural creative hotels. However, there are several areas that could benefit from further elaboration, clarification, or improvement:
1) While the article mentions adopting a qualitative research approach, it would be helpful to provide more details on the specific methods used for data collection and analysis. How were cultural creative hotels selected for study? What criteria were used to identify key factors driving social innovation? 

AR: Thanks for your suggestions. In this revision, we have improved the data collection and analysis, also the criteria to select the two cases in section 3.3.

2)  The article discusses the process of value co-creation in the social innovation ecosystem of cultural creative hotels but could benefit from clearer definitions and conceptualization of terms such as “social innovation,” “value co-creation,” and “innovation ecosystem.” Clarifying these concepts would help readers better understand the theoretical framework guiding the study.

AR: Thanks for your suggestions. In this revision, we have added these terms’ or concept’s definitions.

3) The article proposes practical recommendations for corporate organizations in the field of social innovation, but it could provide more specific and actionable suggestions. How can cultural creative hotels effectively implement the three key elements of the social innovation ecosystem identified in the study? Offering practical guidance would increase the relevance and applicability of the research findings.

AR: Thanks for your suggestions. Thanks for your suggestions. In this revision, we have provided more specific and actionable suggestions. In section 5.1.1, we added how the cultural creative hotels effectively implement the three key elements of the social innovation ecosystem identified.

4) The authors need to engage in more discussion with previous literature in the conclusion section to highlight the theoretical value of this study. Overall, I feel that the theoretical value of this article is low.

AR: Thanks for your suggestions. We have highlighted three value co-creation operation strategies based on the three dimensions of social innovation: (1)Co-evolution of service ecosystem; (2) Initiation of service platform construction; (3) Establishment of collaborative participation in value co-creation for value activities. (in 5.2.1 Social innovation ecosystem and value co-creation). In addition, this article summarizes the connotation and characteristics of the hotel's value co-creation in the innovative social ecosystem (in section 5.2.2.). Third, we have added the explanations of these hotel Exploring the key elements driving the social innovation ecosystem (in section 5.3). Fourth, we added three insights to reinforce the scope of application we proposed by (in 6.1 Conclusion).

Round 2

Reviewer 2 Report

Comments and Suggestions for Authors

The authors responded well to my concerns. Everything is well described and easy to understand. The main conclusions are well-analyzed and derived based on the research findings. This also applies to the theoretical and practical implications.

Comments on the Quality of English Language

NA